# Investigating the Electromechanical Properties of Carbon Black-Based Conductive Polymer Composites via Stochastic Modeling

**DOI:** 10.3390/nano13101641

**Published:** 2023-05-14

**Authors:** Tyler Albright, Jared Hobeck

**Affiliations:** Alan Levin Department of Mechanical & Nuclear Engineering, Kansas State University, Manhattan, KS 66506, USA

**Keywords:** nanocomposite, carbon black, piezoresistivity, electrical conductivity, dispersion

## Abstract

Conductive polymer composites (CPCs) have shown potential for structural health monitoring applications based on repeated findings of irreversible transducer electromechanical property change due to fatigue. In this research, a high-fidelity stochastic modeling framework is explored for predicting the electromechanical properties of spherical element-based CPC materials at bulk scales. CPC dogbone specimens are manufactured via casting and their electromechanical properties are characterized via uniaxial tensile testing. Model parameter tuning, demonstrated in previous works, is deployed for improved simulation fidelity. Modeled predictions are found in agreement with experimental results and compared to predictions from a popular analytical model in the literature.

## 1. Introduction

In a futuristic world, engineering materials will be able to withstand any cyclic loading, changes in environmental stresses, and most traumatic damaging events. These advanced structures will fail only in limited and the most extreme circumstances. In the event of these catastrophic failures, how will we know that the structure has been compromised? How will the health of a structure be assessed and realized prior to its inevitable ultimate structural failure? These conceptual questions can also be asked of composite structural designs of the present. Modern research is focusing on the detection of composite defects and the determination of their influence on the strength and life expectancy of the structure. This field of research is commonly referred to as structural health monitoring (SHM). SHM research aims to supplant design overcompensations and uncertainty with informed analytics on the integrity of the structure in question without compromising the structure via obtrusive sensor networks.

The electromechanical characteristics of carbon fibers (CF) have been exploited for SHM and strain measurement in fiber-reinforced polymer composite (FRPC) structures. Piezoresistivity is defined as the resistivity change in a material when mechanical strain is applied. Wang and Chung observed this phenomenon in a 24-lamina quasi-isotropic carbon FRPC layup. They found that compressing the structure resulted in measurable through-thickness resistivity change in both reversible and irreversible forms [1]. Angelidis et al. demonstrated a method of detecting delamination in a carbon FRPC by measuring laminate piezoresistivity before and after impact damage through arrays of individual electrodes at opposite ends of the laminate [2]. Swait et al. also demonstrated the ability to detect delamination via through-thickness electrical potential analysis of a carbon FRPC panel following a controlled impact event [3].

In contrast to self-sensing conductive FRPC, which rely solely on the load-bearing constituents of the composite, piezoresistive FRPC have been achieved by incorporating conductive filler particulates, with sizes ranging from millimeters to nanometers, into composites with electrically insulating constituents (i.e., glass-fiber composites, epoxies) [4]. These material systems, referred to as conductive polymer composites (CPCs), have since garnered interest in the SHM community as a material class for strain and fatigue transducer development. The benefits of such transducers include the simplicity of their design, ease of their manufacture, and their atomic compatibility/similarity to that of structural composite designs, namely carbon-based FRPC.

Piezoresistivity has been demonstrated in a variety of CPCs with conductive constituents. The most commonly selected conductive filler materials for piezoresistive CPC transducers are carbon-based (graphene, carbon nanotubes, etc.). Nag-Chowdhury et al. demonstrated the viability of deploying a CPC transducer composed of epoxy and high-aspect ratio carbon nanotubes for SHM applications [5]. The prospect of deploying a network of CPC transducers with intent to monitor conductivity drift during service life presents a unique set of engineering problems. Critical problems include the reproducibility of pristine transducer signatures and the development of models used to correlate changes in transducer signature to changes in structural integrity, current factors of safety, and remaining service life.

Numerous manufacturing methods for creating carbon-based CPC transducers have been demonstrated in the literature, including injection/compression molding [6,7,8,9,10,11,12,13,14,15,16,17], extrusion [18,19], melt-spinning [20], 3D-printing and ink-jetting [15,21,22,23,24,25,26], fumigation coating [27,28], spray-layer-by-layer (sLBL) [5,29], and hot pressing [13,30,31,32] among others [33,34,35]. While various manufacturing methods have been reported in the literature, the results of studies of CPCs with similar constituent materials reveal significant variation in nominal electromechanical properties, as shown in Figure 1. To better understand, predict, and control CPC electromechanical properties, more precise manufacturing processes, process validation techniques, and predictive models are needed.

Electrodes for interrogating CPC electromechanical properties are typically achieved through embedded conductors or external surface probing. For external surface probing, researchers have demonstrated measurement repeatability by deploying custom surface clamped electrodes [7,8,10,33,41], while others have deployed commercial volume resistivity test fixtures, such as the Keithley 8009 [13,15,30]. Instead of deploying mechanically reinforced contacts, researchers have also demonstrated the effectiveness of applying conductive surface coatings, such as sputtered gold or conductive silver paste, to create electrical contacts [9,11,12,16,19,23,26,31,32]. Other researchers have utilized the inherent adhesive properties of the CPC constituents to adhere CPC films to metallic structures with exposed contacts [21,24,27,28]. Alternatively, Nag-Chowdhury et al. demonstrated an embedded CF tow could be used to interrogate CPC conductivity and piezoresistivity [5].

CPC piezoresistive responses to both tensile and compressive strains have been reported in the literature. Voet et al. reported observations of electron tunneling in carbon black systems in compressive states as early as 1965 [42]. This phenomenon is a consequence of particle motion within the CPC inducing rearrangement of conductive pathways, thus varying measured conductivity [43]. For researchers interested in the development of CPC flexible strain gauges, reversibility or recovery of the pristine resistivity when unloaded is desired, but often not realized [26,44,45,46,47]. For researchers interested in CPC SHM applications, understanding the underlying principles behind such irreversible conductivity change in fatigued CPC materials is a fundamental challenge on the cutting edge of the discipline.

### 1.1. Analytical Modeling of CPC Electromechanical Properties

Radzuan et al. have presented a thorough review of models for predicting conductivity of CPCs with respect to their concentration of conductive filler(s) [48]. One of the most commonly deployed analytical models in the literature for describing the conductivity of CPCs is the theory of percolation [49,50]. Carmona et al. developed an analytical model based on the theory of percolation to describe the piezoresistive behaviors of CB CPCs [51]. To address the model’s limitations in predicting the effects of various influencing factors (filler particle diameter, matrix compressive modulus, etc.) on piezoresistance quantitatively, and to explain the time dependence of CPC piezoresistance, Zhang et al. proposed an alternative analytical model [52]. The model proposed by Zhang et al. describes a CPC volume consisting of conductive spherical element chains suspended in an insulating matrix spanning parallel conductive electrodes. The resistance can then be approximated with,
(1)$$R=\frac{\left(L-1\right)R_{m}+LR_{c}}{S}\approx \frac{L\left(R_{m}+R_{c}\right)}{S}$$
where *R* is the resistance of the composite, $R_{m}$ is the resistance between individual particle pairs, $R_{c}$ is the resistance across the length of one particle, L is the number of particles forming one conductive chain, and S is the total number of conductive chains spanning the electrodes. The quantum-mechanical theory of electron tunneling, which explains the flow of electrons between conductors separated by insulating barriers, is used to define the resistance between individual particle pairs. Simmons’ analytical formulations [53] propose that the tunneling current J between atomically similar conductors separated by an insulating barrier at low voltages can be defined as,
(2)$$J=\frac{3\sqrt{2m\varphi }}{2s}{\left(\frac{e}{h}\right)}^{2}V\exp\left(-\frac{4\pi s}{h}\sqrt{2m\varphi }\right)$$
where *h* is Planck’s constant, m and e are the electron mass and charge, respectively, *V* is the applied voltage, *s* is the thickness of the insulating barrier, and *φ* is the height of potential barrier between the conductors. Assuming the effective area of conductance between the particles as a^2^, the effective resistance between particle pairs can be defined as,
(3)$$R_{m}=\frac{V}{a^{2}J}=\frac{8\pi hs}{3a^{2}\gamma e^{2}}\exp(\gamma s)$$
where
(4)$$\gamma =\frac{4\pi }{h}\sqrt{2m\varphi }$$

From this formulation, it is observed that the resistance between particle pairs diverges as the thickness of the insulating barrier separating them grows. The resistance across the length of an individual particle is considered negligible compared to the tunneling resistances assumed to be present in each particle chain, therefore $R_{c}$ ≈ 0. Substituting (3) into (1), the resistance of such CPCs can be calculated theoretically as,
(5)$$R=\frac{L}{S}\left[\frac{8\pi hs}{3a^{2}\gamma e^{2}}\exp\left(\gamma s\right)\right]$$

Adapting the model to describe CPC piezoresistive behavior, it is assumed that the separation between conductive particles changes from $s_{0}$ to s when strained. The relative resistance (*R*/$R_{0}$) can then be calculated as,
(6)$$\frac{R}{R_{0}}=\frac{s}{s_{0}}\exp\left[-\gamma \left(s_{0}-s\right)\right]$$
where $R_{0}$ is the CPC’s nominal resistance, and $s_{0}$ is the original assumed interparticle separation. Assuming that the change of interparticle separation along the conducting path is only due to matrix deformation, the separation s can be calculated as,
(7)$$s=s_{0}\left(1-\epsilon \right)$$
where *ϵ* is the strain of the matrix. By assuming the spherical particles to be monodisperse and arranged in a cubic lattice, $s_{0}$ can be calculated as,
(8)$$s_{0}=D\left[{\left(\frac{\pi }{6}\right)}^{\frac{1}{3}}\theta^{-\frac{1}{3}}-1\right]$$
where *D* is the particle diameter, and *θ* is the filler volume fraction of the CPC. Substituting (8) and (7) into (6), the relative change of resistance is defined as,
(9)$$\frac{R}{R_{0}}=\left(1-\epsilon \right)\exp\left\{-\gamma D\left[{\left(\frac{\pi }{6}\right)}^{\frac{1}{3}}\theta^{-\frac{1}{3}}-1\right]\epsilon \right\}$$

This formulation has been modified in the literature for special cases and other considerations such as the effects of creep, excitation voltage, and electrode contact resistance on CPC conductivity [54,55,56,57]. While some modeled assumptions such as all primary particles as perfect spheres and monodispersed have been shown to be untrue by a vast majority of studies in the literature, such assumptions greatly simplify the analytical formulation. These assumptions are also often applied in the stochastic modeling methods discussed in the following section.

### 1.2. Stochastic Modeling of CPC Piezoresistivity

Good agreement between fitted analytical models and experimentally derived CPC conductivity and piezoresistivity data is common. It is far less common to find agreement between reported values of model parameters (i.e., percolation threshold, critical power law exponent, etc.) fitted for experimental datasets from CPCs of nearly identical composition. It is hypothesized that this is a consequence of the stochastic nature of CPC materials and the influence of factors that can be difficult to quantify, such as filler conductivity, the nature of the polymer-filler interactions, processing techniques, and so on.

Recent research has shown the potential of using stochastic modeling methods in estimating CPC characteristics [58,59,60,61,62,63,64,65,66]. These methods utilize the capabilities of high-performance computing systems to create representative volume elements (RVEs) composed of conductive particles suspended in a polymer that is not discretely modeled. A stochastic modeling program was developed for purposes of studying the electromechanical properties of CB-based CPC materials [67]. Experimental results in the authors’ previous research garnered confidence in the ability of the model to accurately represent physical CPC specimens through optical microscopy-based model parameter tuning methods [68].

Strain can be simulated on CPC RVEs by applying directional scaling factors to the interparticle distance calculations of the simulated random resistive networks. These scaling factors are dependent on the physical parameters of the simulation: namely the magnitude and direction of the strain and the Poisson’s ratio of the suspending epoxy. Considering the vast majority of the RVE is composed of isotropic epoxy, the RVE is assumed to exhibit isotropic behavior under stress. Uniaxial tension is simulated in the direction of net current flow through the RVE (x-direction). The RVE is subjected to simulated contraction in the y and z directions. The general relationship between all principal strains can be expressed as,
(10)$$\epsilon_{y}=\epsilon_{z}=-\nu \epsilon_{x}$$
where *ε* is the strain, *ν* is Poisson’s ratio, and subscripts denote the principal direction associated with each strain. Various magnitudes of global uniaxial tensile strain are applied to the RVE. Local strain values are then calculated and applied to the interparticle distance calculations. New interparticle relationships are defined, and a modified resistive network is formed where the resulting RVE resistance is calculated via the methods described in the authors’ previous research [6,67].

In this research, the stochastic modeling program developed by the authors in previous works [68] is used to generate predictions of CPC electromechanical properties. A simplified formulation of Simmons’ theory of tunneling conductivity is deployed to study the conductivity and piezoresistivity of CB-based CPC specimens that are compared directly to physical CPC specimens of whom the stochastically generated volumes are modeled. The proposed modeling and testing framework captures the nanoscale, macroscale, stochastic, and fractal features of CB CPC systems, and presents a novel technique for accelerated selection and optimization of CPC constituents for engineering applications.

## 2. Materials and Methods

Casting was identified as a cost effective, simple, and repeatable manufacturing method for molding room-temperature curable CPC transducers. To create casting molds, a room-temperature cure two-part silicone epoxy was poured into 3D printed molds. An image of the 3D printed inverse dogbone-shaped mold is shown in Figure 2a. This 3D printed structure is used to create the silicone dogbone-shaped molds shown in Figure 2b,c, which are then used to cast dogbone-shaped CPC samples, such as the sample shown in Figure 2d. The shape of the sample was designed in accordance with the ASTM D638-14 standard [69]. Slots were created in the silicon mold to allow the placement of CF tows in precise locations about the sample gauge length. Further, 3 k CF tows were placed in the slots prior to the deposition of the CPC to limit the flow of the CPC into the ends/tabs of the mold. The sample tabs were poured with neat resin to eliminate the influence of conductive pathway formation via the metallic grips of the hydraulic load frame.

A master batch was prepared by first incorporating measured quantities of CB (Vulcan ^®^ XCMax22, Cabot Corp.) and resin (EZ-Lam, ACP Composites) via hand stirring, followed by shear mixing using a 3 roll mill (Torrey-Hills). The content of CB in the master batch was controlled to produce cured CPC samples containing approximately 3.15% CB by weight. Quantities of the master batch were isolated in a beaker and a thinning agent (200 proof ethanol) was added at a ratio of 1 part thinner to 1 part resin. Ultrasonic probe (Boshi Electronic) homogenization was performed at 65% of maximum power (40 Watts) at a duty cycle of 40%. After homogenization, the mixture was baked in a conventional oven at 82 degrees Celsius for 96 h with occasional stirring to evaporate the vast majority of the ethanol. After evaporating the solvent and allowing the mixture to cool to room temperature, the hardening agent was incorporated into the final mixture via hand stirring. Finally, the mixture was poured into the casting mold, and excess resin was scraped from the mold using a straight edge tool. The samples were allowed to cure for 48 h at room temperature. After removal of the samples from the cast, the edges of the sample were trimmed using a razor blade.

To measure the change in sample resistance during strain deformation, the CF electrodes were mechanically connected to lead-wires using metallic alligator clips. A CompactDAQ (cDAQ), paired with LabView (National Instruments), was used to record measurements of sample resistance during testing. The cDAQ was equipped with a strain/bridge input module (NI-9237) and a RJ-50 to the screw-terminal connector (NI 9949) to streamline the data acquisition process. Connecting the transducer in a quarter-bridge configuration required a precision resistor to be supplied to match the nominal resistance of the transducer. The CPC nominal resistances ranged from 100 kΩ to 500 kΩ. To cover this range of resistances, multiple high-resistance potentiometers were soldered together in series on a prototype breadboard and were subsequently connected to the bridge module. By connecting potentiometers to the bridge, the bridge could be balanced without the need for programmatic compensation. Once the bridge was balanced successfully, the piezoresistivity of the sample could be examined.

## 3. Results

Measurements of pristine sample geometry and resistance were acquired prior to uniaxial tensile testing. The average length, width, and thickness of the gauge length region of the CPC samples were 56 mm, 13 mm, and 3.8 mm, respectively. Mixtures were produced with ultrasonic homogenization energies 50 kJ and 100 kJ. These mixtures were spin coated and imaged via light microscopy to investigate dispersion characteristics via the methods described in the authors’ previous research [68]. Images from the spin-coated films are shown in Figure 3a,b. The mixture prepared with 100 kJ of homogenization energy was considered to be of the greatest possible dispersion via the processing techniques deployed. The effects of phase separation during cast sample curing on the agglomeration of CB primary aggregates is beyond the scope of this study, and such effects are assumed to be negligible for purposes of stochastic model simplification. Additionally, the modeled CB primary particles are assumed to be monodispersed and perfectly spherical to simplify stochastic volume generation and analysis. Measurements from cast CPC samples prepared with the 100 kJ mixture were to be compared directly to stochastic model predictions. The average and standard deviation of ten 100 kJ mixture energy CPC samples’ resistivity at a pristine state was 0.261 Ω-m and 0.085 Ω-m, respectively. The average and standard deviation of six 50 kJ mixture energy samples’ resistivity at a pristine state was 1.940 Ω-m and 0.249 Ω-m, respectively.

Individual CPC dogbone samples were carefully secured in the load frame grips under a no-load condition, and uniaxial strain tests were performed while monitoring the change in sample resistance using the aforementioned quarter-bridge data acquisition setup. In addition to monitoring the CPC electrical properties, the elongation of the sample is measured at the gauge length using an extensometer (Epsilon). Figure 4a shows the extensometer and CPC resistance change data for a sample being strained at a constant rate of 0.2 mm per minute. Since the time-series data from the extensometer and CPC sample are coincident, the relative resistance change can be plotted in relation to the local strain measurement, shown in Figure 4b. At low strains (ϵ < 0.004), the relationship between measured strain and relative change in the sample resistance appears to be linear. The relationship between measured stress and strain, and stress and relative resistance change, also appeared to be correlated linearly. At peak strain, the sample was held for a short time before being unloaded at a constant strain rate equivalent to that of the initial ramp rate. At this point, the absolute value of the slope of the relative resistance change appears to have increased. This is speculated to be the consequence of unrecoverable plastic deformation of the matrix. Such plastic deformation would result in compressive stress if the sample was forced to return to the original undeformed state. Assuming the current carrying network only deformed elastically, it is hypothesized that the compressive stress state would close the distance between neighboring primary particles in a tunneling relationship. The same uniaxial tensile test was performed on each sample and the results of those tests are tabulated in Table 1. The slope of the best-fit line relating relative resistance change to strain is equivalent to a sample’s gauge factor (GF). The average GF, and standard deviation of the measured GFs, of the 100 kJ samples was slightly less than that of the 50 kJ samples. The most significant difference between the sample mixtures was their average resistivity: 0.26 for 100 kJ, and 1.94 for 50 kJ. Also worthy of note, the average elastic modulus of the 100 kJ CPC samples was 16.95 GPa. For comparison, the average elastic modulus of a set of six neat polymer dogbone samples was 5.78 GPa.

### Comparison of Predictive Models

Stochastic CPC RVEs were generated with the aforementioned fractal aggregate-based stochastic modeling program. A set of CPC simulation parameters were maintained for the generation of high-fidelity RVEs, defined in accordance with the results of the model fidelity validation and tuning research presented in a previous article [68]. Relevant simulation parameters are as follows: Number of primary particles per fractal aggregate = 100, fractal dimension and prefactor of fractal aggregates = 2.7 and 0.7, diameter of primary particles = 30 nm, weight percentage of CB in CPC = 3.15%, percentage of aggregate particles placed at random = 0.028%, percentage of particles placed via aggregate forced agglomeration = 20%, maximum allowable degree of interparticle penetration = 5 nm, maximum tunneling distance considered numerically relevant = 3 nm. The convergence of the model’s conductivity predictions with respect to RVE cubic length is shown in Figure 5c. This trend is consistent with previously demonstrated simulation convergence behaviors observed with RVEs of perfect dispersion [67]. In contrast to the perfect dispersion convergence threshold, which was estimated at a RVE cubic length ≈ 6200 nm, convergence was estimated for this set of simulation parameters at a RVE cubic length ≈ 12,000 nm.

Hundreds of RVEs, of cubic length of 12,000 nm, were subjected to incremental uniaxial strain, and the relative change in RVE resistance at each increment of strain was calculated following the method discussed in Section 1.2. The polymer barrier height simulation parameter (λ), corresponding to the electron tunneling barrier height of the insulating matrix between tunneling pa, was varied to study the influence of the parameter on the predicted resistance change of the simulated RVEs. Each RVE relative resistance change is calculated for varying values of λ, and the individual data points and averages are plotted in Figure 5a. Evident in Figure 5a, varying the assumed value of λ influences the rate of relative resistance change with respect to strain. In other words, the assumed value for λ directly influences the GF predictions of the stochastic model.

The simulation parameter defining the resistance between particle pairs in direct contact ($R_{c}$) was also varied to study its influence on the predictions of conductivity for RVEs at zero strain. After searching the literature and finding no experimentally derived values, an arbitrary value for $R_{c}$ (1000 Ω) was used in initial simulations. The same RVEs used to generate the data in Figure 5a were used to generate predictions of unstrained resistivity while varying $R_{c}$. The results are shown in Figure 5b. As the value of $R_{c}$ is increased, the average resistivity of the RVEs is increased. The standard deviation of the average resistivity was also found to increase. The average measured resistivity of the experimental samples is plotted in Figure 5b for reference. These results show that by varying λ and $R_{c}$ within physically feasible bounds, the predictions of the electromechanical properties of the modeled CPCs can be tuned to more closely resemble characteristics of physical CPC systems.

Piezoresistivity predictions for the cast CPC samples were computed using the analytical model developed by Zhang et al. [52], detailed in Section 1.1. The parameters in the analytical model were defined according to the specifications of the cast CPC experimental samples. The analytical predictions of Zhang’s model are compared to experimental results from a single CPC specimen in Figure 6. As evident from this plot, major disagreement exists between the experimental results and the original model predictions. The original derivation of the model by Zhang assumes the spherical conductive filler in the CPC is arranged in a cubic lattice for simplification of the mathematical calculation of the average interparticle tunneling distance ($s_{0}$) [52]. The research previously presented by the authors provides evidence that this assumption is not appropriate for real mixtures due to their stochastic particle dispersions [68]. Instead, the cubic lattice assumption serves as a theoretical upper bound for such CPC systems with maximally ordered particle distributions of isotropic interparticle spacing ($s_{0}$). Straying away from the cubic lattice assumption proposed in the original model derivation, predictions of piezoresistivity were generated considering a range of values for the variable $s_{0}$. The polymer barrier height variable for the analytical model was set at 0.5 eV, consistent with ranges reported in the literature for similar polymers [59]. Good agreement between experimental and analytical predictions occurred when $s_{0}$ was set to a value between 0.5 and 1.0 nm, as shown in Figure 6. For comparison, the idealized cubic lattice assumption resulted in a value of $s_{0}$ ≈ 15 nm.

Figure 6 plots the predictions of the stochastic and analytical models alongside the experimental data from low-strain portions of a uniaxial tensile test of a cast CPC sample with electromechanical properties near the average of the sample set. From this plot, it is evident that the stochastic model results exhibit gauge factors comparable to the physical CPC specimens at low strains. These results are derived from the application of widely accepted physics-based models to stochastically generated RVE systems with agglomeration features optically comparable to those observed in experiment. The physical model parameters λ and $R_{c}$ are varied within physically feasible and published ranges defined in the literature, and agreement is found between both predictions of nominal conductivity and gauge factor. Additionally, the deviations of pristine resistivity and gauge factor observed in experimental samples are comparable to the deviations of the same predicted properties obtained via the stochastic model, graphically illustrated in Figure 5a,b. Such deviations in nominal properties are indicative of the stochasticity of the physical specimens. The stochastic nature of the physical specimens is captured via the stochastic modeling algorithm, and agreement between the model’s property predictions and experimental results can be tuned by varying simulation parameters representing physical phenomena that are difficult to measure directly (λ and $R_{c}$) but remain within physically feasible limits.

## 4. Conclusions

The research presented in this paper supports claims that the electromechanical behaviors of CPC materials can be accurately described at various length scales via the stochastic modeling approach. By deploying an experimental data-driven model-parameter tuned stochastic modeling approach, vast numbers of spherical element-based CPC configurations can be numerically explored in a timely manner. A casting method was developed for manufacturing CPC specimens with precise geometries and consistent electrode conditions for comparison of experimentally derived electromechanical properties to analytical and stochastic model predictions. In addition, 3D printed structures were used to create flexible molds out of silicone epoxy. The resulting molds were used to create dogbone-shaped specimens consistent with the specifications of ASTM D638-14. The molds were designed with slots for CF tows to be embedded in parallel about the gauge length of the dogbone-shaped specimens. CPC samples composed of 3.15% by weight CB were cast using the flexible molds and subjected to uniaxial strain tests. The average relative change of sample resistance with respect to strain for all samples was calculated and compared to predictions of the stochastic model and a popular analytical model from the literature. Modifications to the analytical model are presented.

The experimental characterization of CPC dogbone-shaped specimens presented in this research was limited to simple uniaxial tensile tests to determine piezoresistive characteristics for stochastic model tuning. During such experiments, non-recoverable strain deformation and creep behaviors were observed. These phenomena were accompanied by permanent changes in CPC conductivity and piezoresistivity. Understanding the sources of influence for these permanent changes in CPC conductive networks is crucial for the development of fatigue sensing CPC transducers. The high-fidelity RVEs generated by the stochastic model provide a direct approach to describe the causes of such phenomena. In future works, simulated RVEs should be adapted for finite element and micromechanics investigations of CPC plastic deformation and creep. Additional experimental efforts for quantifying fatigue influence on the electromechanical properties of CB CPCs will be necessary to support such numerical studies.

## Figures and Tables

**Figure 1 nanomaterials-13-01641-f001:**
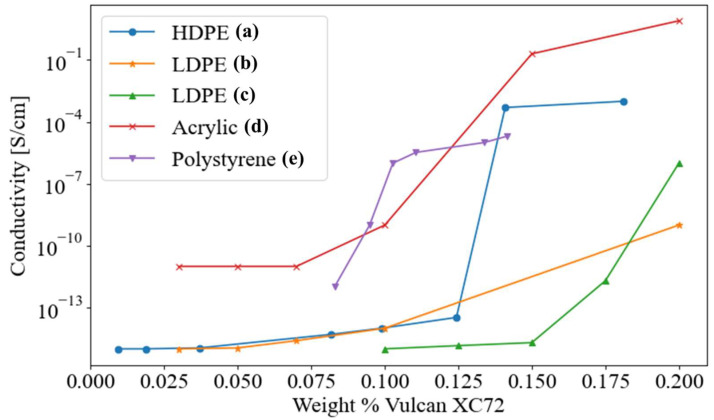
Plot of the CB-based (Vulcan XC72, Cabot Co.) CPC conductivities with varying polymer matrices reported by various sources: (**a**) [36]; (**b**) [37]; (**c**) [38]; (**d**) [39]; (**e**) [40].

**Figure 2 nanomaterials-13-01641-f002:**
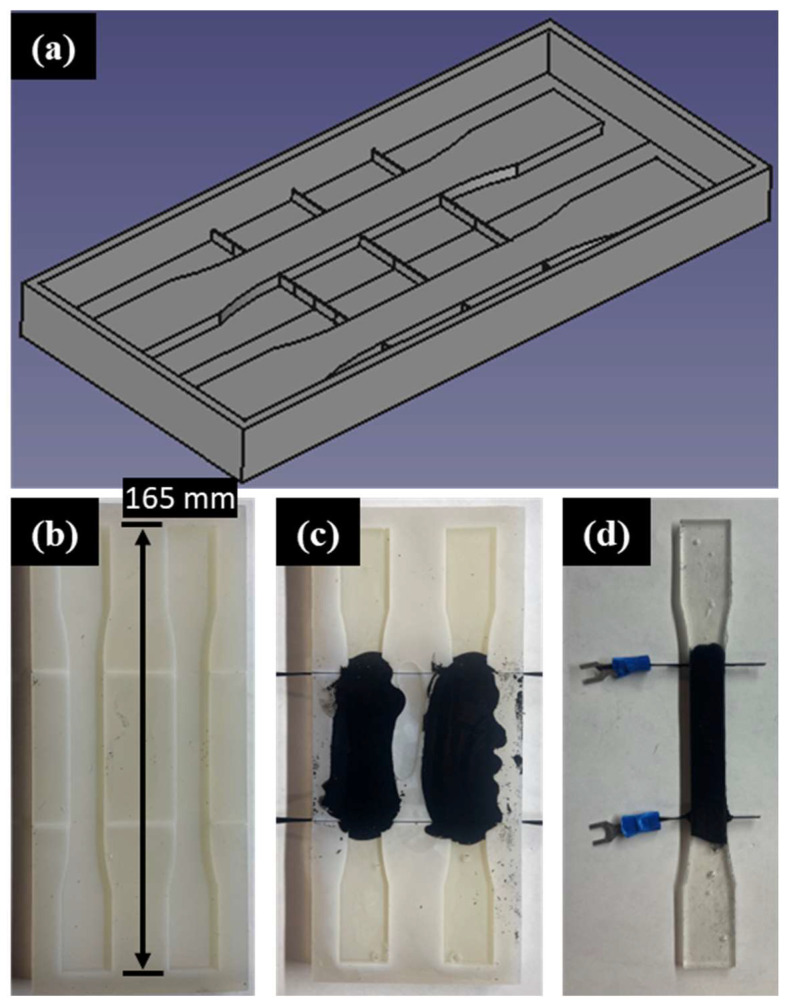
(**a**) Rendering of 3D printed dogbone mold maker; (**b**) Cured silicone epoxy dogbone mold; (**c**) Dogbone mold with CPC transducer curing; (**d**) CPC dogbone sample with easy-to-access electrical leads ready to be tested.

**Figure 3 nanomaterials-13-01641-f003:**
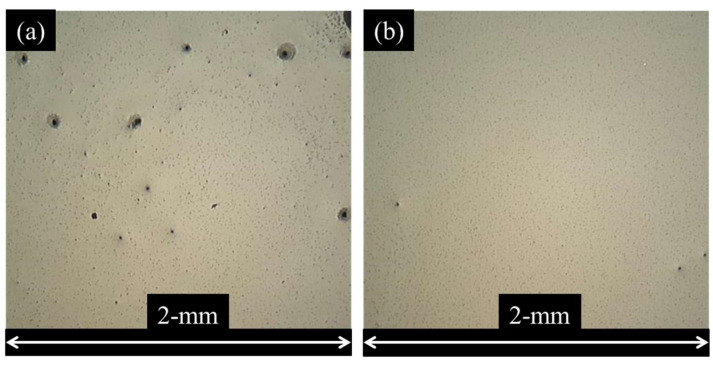
Images of cured spin coated CPC (3.15% by weight CB) film samples (diluted 1:1 ethanol to resin) with homogenization energies of (**a**) 50 kJ and (**b**) 100 kJ.

**Figure 4 nanomaterials-13-01641-f004:**
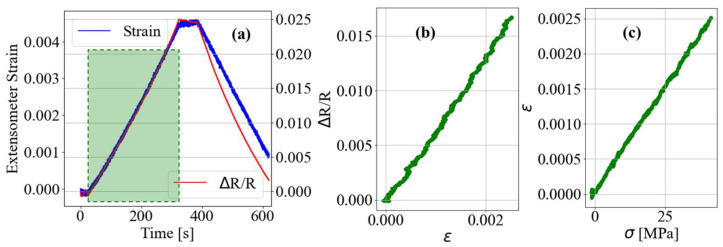
(**a**) Plot of extensometer and CPC sample resistance measurements obtained undergoing a uniaxial tensile event with incremental displacement control rate of 0.1 mm per minute; (**b**) Plot of relative CPC sample resistance change with respect to measured local strain; (**c**) Plot of strain with respect to stress for single CPC sample undergoing uniaxial strain elongation event.

**Figure 5 nanomaterials-13-01641-f005:**
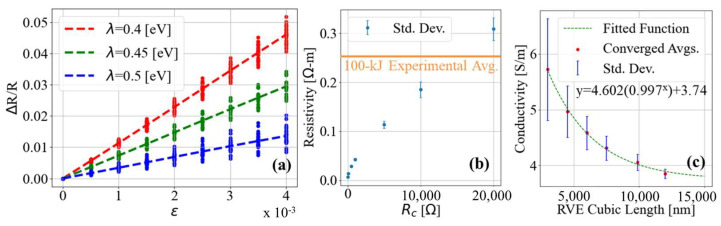
(**a**) Plot of average relative resistance change of RVEs subjected to uniaxial strain while varying values of polymer barrier height, a parameter used in calculating tunneling resistance between simulated particle pairs. (**b**) Plot of nominal resistivity predictions for simulated RVEs considering varying values of contact resistance between particle pairs in direct contact. (**c**) Plot of converging predicted conductivity with respect to RVE cubic length.

**Figure 6 nanomaterials-13-01641-f006:**
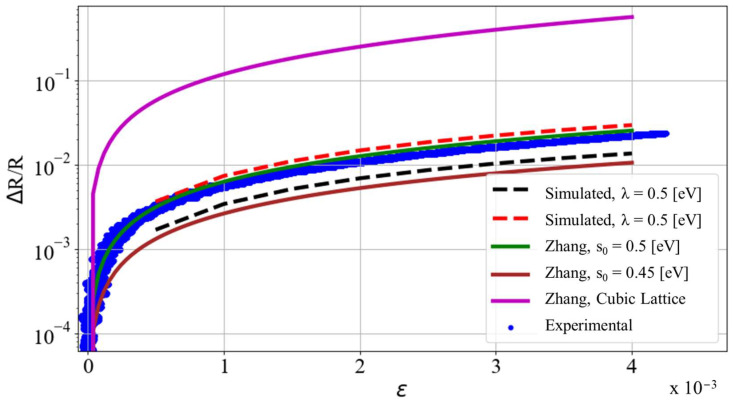
Plot of relative change of resistance with respect to strain for experimental CPC samples undergoing uniaxial tensile strain events of varying displacement rates, simulated RVEs with varying assumed polymer barrier heights (*λ*), and analytical predictions from Zhang’s piezoresistivity model with varying assumed pristine state average tunneling distances ($s_{0}$).

**Table 1 nanomaterials-13-01641-t001:** Tabulated experimental data from uniaxial tensile tests of cast CPC/neat-polymer specimens of varying dispersion characteristics.

Mix Energy		ΔR (ε)	ΔR (σ)	σ (ε) ≈ 1/E [MPa^−1^]	R_Pristine_ [kΩ]
100 kJ	Average	8.010	4.69 × 10^−4^	5.90 × 10^−5^	296.17
	Std. Dev.	0.792	5.62 × 10^−5^	9.20 × 10^−6^	96.10
	Std. Dev. (%)	9.88%	12.00%	15.57%	32.45%
50 kJ	Average	8.566	5.38 × 10^−4^	6.50 × 10^−5^	2200
	Std. Dev.	1.372	2.59 × 10^−5^	1.45 × 10^−5^	281.0
	Std. Dev. (%)	16.02%	4.82%	22.32%	12.77%
Neat	Average	-	-	1.73 × 10^−4^	-
	Std. Dev.	-	-	1.88 × 10^−5^	-
	Std. Dev. (%)	-	-	10.88%	-

## Data Availability

The source code developed by the authors and deployed to generate the stochastic model data presented in this paper has been made publicly available in the following GitHub repository: https://github.com/tylerbalbright/Stochastic_Model_CB_FracVal.

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
