# Peer review of "Investigating the Electromechanical Properties of Carbon Black-Based Conductive Polymer Composites via Stochastic Modeling"

_nanomaterials, 2023, doi:10.3390/nano13101641_

Round 1

Reviewer 1 Report

General comments:

I enjoyed reading this paper and I thank the authors for the opportunity to review their work. I feel it is worthy of publication. There are a number of issues that I think need to be resolved. Much of the results support the hypothesis that electron tunnelling controls the resistance in nanocomposites. The key result here is that the stochastic model makes a much better prediction than the cubic lattice. However, the issue is that the experimental work has a large number of aggregates that drops the percolation threshold considerably, and that requires the inclusion of a large number of adjustable parameters into the fit. That leads to the question of the sensitivity of the model on the adjustable parameters. The important one being the degree of aggregation this essentially sets the percolation limit and the scaling on the resistance. I would like to see some sensitivity analysis performed on this parameter. This would allow the reader to determine the value in this approach. The degree of aggregation is difficult to determine experimentally and I feel it would be outside the scope of this work. 

More focused comments:

Within the introduction, please make a more direct link to the work of Voet et al. (1965) https://link.springer.com/article/10.1007/bf01497077

On electron tunnelling in carbon black. While this is a study of carbon black under pressure it does establish electron tunnelling as the mechanism.

(and/or the meeting abstract in Carbon 1964 (1),3 p388 Van Beek and Van Pul)

Optical microscope image is not high enough magnification to show the aggregates that still exist in the material. A 3.15% percolation threshold would indicate a high degree of aggregation. The percolation limit for spherical particles is 28%.  Curing also leads to aggregation, via phase separation, so showing a spin cast film, (where the solution may have been further diluted to allow for spin casting) and assuming the CB is the same in both is misleading. I presume the solution for the spin coating was the 1:1 ethanol:epoxy mixture? Could you please make this clear in the text and caption for figure 3.

Line 130 The assumption that the particles are spherical and monodisperse, is clearly incorrect in the vast majority of studies of carbon black dispersions. However, this does not mean this analysis is not worthwhile and therefore I would suggest that the authors better explain this in the introduction to section 1.2.

Line 176 C should be lowercase

Line 191 please convert to Celcius

Images in figure 2 need scale bars.

Figure 4: The resistance follows the strain on extension but not compression. This would indicate that these two processes are different. I think this is worth mentioning even if you can only speculate on the reason.

Table 1: and text line 225-226: can you check the resistance and resistivity values? I do not get the same numbers as you have.

The model does include a large fraction of aggregated particles:

Line 268: “percentage of particles placed via aggregate forced agglomeration – 20%”

Does this indicate that 20% of your 30 nm CB particles are in an aggregate of some form with a contact resistance of Rc (1000 ohms) between the CB particles? [100 of these in a sphere would have a radius of 139nm and a resistance of circa 4.6 kohms.]

What does “The polymer barrier height simulation parameter (λ)” actually mean as a physical parameter, is this an electron tunnelling barrier height? 

I think it's pretty obvious that increasing the polymer barrier height simulation parameter which controls the resistance between the particles, will increase the slope of the resistance versus strain. This is afterall just a reciprocal scalar multiplier on the separation-resistance value.  The higher the jump barrier the more difficult tunnelling will be and therefore the less likely electrons are to tunnel and therefore the current goes down at a fixed voltage, analogous to a higher resistance. This makes me feel you don’t truly understand what the simulation is doing or that you cannot find a worthy conclusion from the simulation.

Again this is illustrated in figure 6, as a direct consequence of electron tunnelling.

The referencing needs attention, for example reference 50 is incorrect “Carmona et al” in the text but “Kanoun, O.; Bouhamed, A.; Ramalingame, R.; Bautista-Quijano, J.R.; Rajendran, D.; Al-Hamry” in the references. Carmona does not appear anywhere in the reference list.

Reviewer 2 Report

This paper presents a report on conductive polymer composites (CPC), through a combination of experimental and theoretical methodology.  The paper is well-written and presented, yet with some minor aspects on the figures that could be tackled.  Despite the extensive introduction, the scope/novelty of the research is not clear, and the main research outcome seems to rely on the stochasticity of the physical samples.  The preparation method seems quite manual (thought interesting), and potentially practical for future investigations.  All materials are commercial, and there is not a clear strategy on composition.  There is a lack of physic-chemical analysis on the specimens (FT-IR, TGA?), which could be beneficial to assess the results.  The experimental response of dogbone-shaped specimens presented was limited to simple uniaxial tensile tests, without a proper dynamic analysis (what is the effect of frequency and temperature on the response).  As the authors mention, there are still some very relevant tests/properties (micromechanics investigations, plastic deformations) that may be needed to compile a complete piece of work.  In short, even though the work has engineering merit, the experimental conditions/tests limit its scientific impact, since it is not possible to establish proper structure/properties relationships.  

Reviewer 3 Report

Dear Authors,

Thank you for submitting your manuscript entitled "Stochastic Modeling of Carbon Black Reinforced Cementitious Composites for Electromechanical Sensing Applications" to tha Nanomaterials. After careful evaluation, we regret to inform you that we are unable to accept your manuscript for publication in its current form.

Our primary concern is that the manuscript lacks sufficient experimental investigations on electrochemical and cyclability properties, which are crucial for electromechanical sensing applications. Additionally, the characterization of the CPC dogbone-shaped specimens presented in the research was limited to simple uniaxial tensile tests to determine piezoresistive characteristics for stochastic model tuning. During such experiments, non-recoverable strain deformation and creep behaviors were observed, which were accompanied by permanent changes in CPC conductivity and piezoresistivity. These phenomena need to be thoroughly investigated and understood for the development of fatigue sensing CPC transducers.

Moreover, we suggest that the authors should consider further refining the analytical model presented in the manuscript to better match the experimental results. Additionally, it would be beneficial to explore the finite element and micromechanics investigations of CPC plastic deformation and creep using the high-fidelity RVEs generated by the stochastic model.

We appreciate the time and effort that you have put into this manuscript and encourage you to revise and resubmit it with the necessary improvements. If you choose to resubmit, we will do our best to provide a timely and constructive review.

Reviewer 4 Report

A manuscript contains original research work where the authors have presented the results of the study of the electromechanical properties of conductive polymer composites (CPC) based on epoxy resin and 3,15 % carbon black. The autors very successfully combined theoretical process modeling approaches and experimental studies of piezoresistive parameters of uniaxial stretching of dog-bone-shaped samples. The theoreticaly predicted results obtained using a high precision scholastic model are in complete agreement with the experimental data. The result of the work seem to be highly demanded in the development fatigue sensing CPC transducers. This work has prospectives for continuation in terms of quantitative evaluation of fatigue effect on electromechanical properties of CPC. Overall, I highly appreciate this work and believe that it can be published as presented.

Author Response

Thank you for reviewing our article.